# Inorganic Nanoparticles Applied for Active Targeted Photodynamic Therapy of Breast Cancer

**DOI:** 10.3390/pharmaceutics13030296

**Published:** 2021-02-24

**Authors:** Hanieh Montaseri, Cherie Ann Kruger, Heidi Abrahamse

**Affiliations:** Laser Research Centre, Faculty of Health Sciences, University of Johannesburg, P.O. Box 17011, Doornfontein 2028, South Africa; montaseri.hanieh@gmail.com (H.M.); habrahamse@uj.ac.za (H.A.)

**Keywords:** Breast cancer treatment, Photodynamic therapy, Inorganic nanoparticles, Active targeting

## Abstract

Photodynamic therapy (PDT) is an alternative modality to conventional cancer treatment, whereby a specific wavelength of light is applied to a targeted tumor, which has either a photosensitizer or photochemotherapeutic agent localized within it. This light activates the photosensitizer in the presence of molecular oxygen to produce phototoxic species, which in turn obliterate cancer cells. The incidence rate of breast cancer (BC) is regularly growing among women, which are currently being treated with methods, such as chemotherapy, radiotherapy, and surgery. These conventional treatment methods are invasive and often produce unwanted side effects, whereas PDT is more specific and localized method of cancer treatment. The utilization of nanoparticles in PDT has shown great advantages compared to free photosensitizers in terms of solubility, early degradation, and biodistribution, as well as far more effective intercellular penetration and uptake in targeted cancer cells. This review gives an overview of the use of inorganic nanoparticles (NPs), including: gold, magnetic, carbon-based, ceramic, and up-conversion NPs, as well as quantum dots in PDT over the last 10 years (2009 to 2019), with a particular focus on the active targeting strategies for the PDT treatment of BC.

## 1. Introduction

Breast cancer (BC) is the most prevalent malignancy among women worldwide [1]. Cisplatin (cis-diammine-dichloroplatinum (II)) is currently an approved drug that can be utilized for the treatment of various cancers, since it inhibits DNA replication and chain elongation [1]. Although numerous anticancer drugs have been developed over the years for BC treatment, it still remains a therapeutic challenge; since BC can metastasis, become resistant to certain drugs, as well as exhibit lesions of recurrence after surgery [2]. More importantly, conventional anticancer drugs, when administered, spread throughout the body, and thus affect healthy cells and tissues, instead of just the localized tumor area, which requires treatment [3].

The effectiveness of photodynamic therapy (PDT) in ablating localized BC tumors, with limited side-effects is a significant breakthrough in unconventional treatments [4]. PDT can be performed as an adjuvant to other therapies, since it enables selective, as well as localized damage to tumors and their surrounding vasculature [4]. PDT is based on the activation of a nontoxic photosensitizer (PS) with an appropriate light to produce reactive oxygen species (ROS), which in turn eradicates cancer cells [4]. However, due to the hydrophobic nature of most PSs, they have high tendency to aggregate in aqueous solution, reducing the efficacy of PDT treatment [5]. In addition, PSs do not tend to bind to tumor cells selectively, resulting in poor specificity uptake in cancer cells and so localized normal tissues can become affected during treatment [6]. In this context, the combination of nanotechnology and PDT in the form of nanoplatforms is of great importance, whereby PSs are covalently or non-covalently bound to the nanoparticles (NPs) [6]. The selectivity of the nano delivery agents can be also enhanced using active targeting, whereby antibodies and small ligands can be bound to NPs, and so allow for PSs to be specifically (as well as directly) delivered into targeted tumor cells [6]. Therefore, the aim of this review is to collate and discuss the types of inorganic NPs that have been used for the active (as well as targeted) delivery of PSs within PDT BC treatment.

## 2. Conventional Treatments of Breast Cancer 

Chemotherapy, surgery, and radiotherapy are the main therapies utilized for small sized BC tumors [7]. Some other less invasive treatments, such as cryotherapy, laser ablation, and radiofrequency ablation (RFA), have also been developed for early stage BC [8,9]. In spite of promising developments in the treatment of early BC, surgery is generally the first option. Often, positive BC tumor margins can remain unresected, and so the possibility of reoccurrence is eminent. Thus, most often, patients require additional surgeries and chemotherapy treatments [10,11]. In this context, new (and far more effective) treatment modalities are sought after in order to mitigate the collateral damage, as well as improve the treatment outcomes of BC.

## 3. Photodynamic Therapy (PDT) and Photosensitizers (PSs)

PDT is an alternative non-invasive therapeutic technique for the treatment of various types of cancers and non-oncological diseases [4]. PDT is painless and its selectivity to cancer cells is well tolerated by patients [12]. It involves three main aspects: (1) a photo active compound or PS that accumulates in neoplastic cells; (2) local light to excite and activate the PS; as well as (3) surrounding tumor molecular oxygen [13]. When a PS becomes activated through illumination at an appropriate wavelength and it reacts with surrounding molecular oxygen, it produces reactive oxygen species (ROS) and singlet oxygen, which destroys localized tumor cells [13].

Among the broad spectrum of light, ultraviolet (UV) light (200–400 nm) may damage biological components, and so its biomedical applications are restrained, while visible light in the range of 400–650 nm can be utilized for activation of various PSs [13]. In addition, “biological transparent windows” in near infrared (NIR)-I (750–1000 nm) and NIR-II window (1000–1700 nm) enjoy low absorption and scattering, with deep tissue penetration and low auto-fluorescence from biological tissues, and so can be utilized for biophotonic imaging [14,15].

Tissue penetration depth of light can be sometimes limiting, which can affect the amount of PS activated, which in turn affects the amount of ROS and singlet oxygen produced to kill tumor cells [16]. Short wavelengths (<650 nm) generally have a lower penetration depth in tissues, while longer wavelength (above 850 nm) ranges are not sufficient enough to excited or activate PSs [16]. Thus, the most appropriate wavelength for PDT is between the range of 600 and 850 nm, which is known as the “phototherapeutic window” (Figure 1) [16].

The most effective PSs in PDT cancer applications are chemically pure and stable, as well as have minimal dark toxicity and side effects, with ideal hydrophilic properties [17]. Additionally, PSs should have strong absorption within the range of 600–850 nm, as to ensure, limited scattering, high tissue depth penetration, with maximum extinction coefficients [17].

Hematoporphyrin derivative (HpD) and photofrin are the first-generation commercial PSs, known for harsh PDT unwanted side effects [18]. While aminolevulinic acid (ALA), esterified derivatives of ALA and phthalocyanine compounds, which are considered second generation PSs, are known to produce minimal PDT side effects, with improved ROS generation, due to their longer absorption wavelengths, with improved tissue depth penetration [18]. Moreover, conjugation of second generation PSs to various biological carriers (such as nanoparticles) are referred to as third generation PSs, since these “carrier” conjugations generally allow PSs to selectively accumulate in cancer cells [19].

Thus, the activation wavelength, solubility, octanol/water partition coefficient, and molar extinction coefficient are of great importance for determining the potency of a PS in PDT. The main parameters of some PSs are compared in Appendix A.

## 4. Mechanisms of Photodynamic Therapy

There are two main PDT mechanisms, which occur in tumor cells, in the presence of molecular oxygen (Figure 2). Upon irradiation of a PS with a wavelength of light coinciding its absorption spectrum, the PS molecule becomes converted from a ground state to a singlet excited state [20]. The PS drug loses a part of its energy through fluorescence and the remainder is transferred and so the singlet state PS becomes excited to a triplet state. In a type I mechanism, the triplet excited state PS interacts with biomolecules from tumors surroundings to form radicals, which react with molecular oxygen to produce ROS, such as hydrogen peroxides, superoxide anion radicals, and hydroxyl radicals [20]. In the type II mechanism, the energy from an excited triple state PS is directly transferred to triplet state oxygen (^3^O_2_) to form singlet oxygen (^1^O_2_) (Figure 2A) [20]. Both ROS and ^1^O_2_ induce cancer tumor cell death via either apoptotic, necrotic, or autophagy cell death pathways, depending on the intracellular localization of a PS [21]. Apoptotic cell death is usually due to mitochondrial PS localization and damage, whereas necrotic cell death is mostly due to cell membrane damage and loss of integrity. Within autophagy cell death, usually the PS induces endoplasmic reticulum or lysosomes damage; however, this form of PDT induced tumor cell death is not favored since cells can recover [21].

## 5. Passive and Active Targeting PS Uptake Strategies

PS subcellular localization uptake can be classified into either passive or specifically active targeting (Figure 2B). Passive PS uptake is encouraged via the permeability and retention (EPR) effect, which causes tumor tissues to present a leaky vasculature [22]. It is natural occurring process, which utilizes the difference in anatomical and pathophysiological abnormalities of cancer tissue versus normal cells to improve PS passivation in tumor cells [22]. When nanoparticle carriers are bound to PS, they tend to promote the passive uptake of PSs via the EPR effect [22]. Active targeting requires the binding of specific targeting ligands, such as antibodies, peptides, aptamers, folic acid (FA), small ligands, or carbohydrates, onto the surface of PS-loaded nanocarrier systems, which are explicitly overexpressed only on cancer cell receptors; thus, PS uptake in these cells is specifically enhanced and actively internalized [22].

In comparison to passive targeting, nanoparticle active targeting most definitely does provide a more selective absorption of PS in tumor cells with improved PS concentration accumulation; thus, higher accumulation of the nanocarrier and cellular concentration of the drug into the cells will take place [22,23].

## 6. Nanoparticle Delivery Systems for PDT

Drug delivery systems based on nanoparticles (NPs) are a promising approach in PDT to enhance PS absorption in cancer cells. A large surface to volume ratio of the NPs can promote the loading capacity of PSs and so improves concentration delivery and either passive or active uptake in cancer cells [24]. In addition, anchoring of PSs to NPs can improve either the stability and solubility, as well as reduce dark toxicity and enhance localized delivery, to improve PDT treatment outcomes and minimize unwanted side effects [24]. Moreover, the small size of NPs, not only assists PSs to accumulate in cancer cells via passive or active targeting, but also allows these nanocarrier to mimic biological molecules and, thus, easily pass through immune system barriers [25]. In relation to active targeting, PS nanocarriers are usually further functionalized with specific ligands, which are compatible to overexpressed tumor sites, to improve their biocompatibility and specific targeted absorption [26].

Various organic and inorganic NP carrier platforms have been developed over the years for improved PS uptake and enhanced PDT treatment of BC. In this current review, the utilization of actively targeted inorganic NPs for PDT of BC has been discussed over 10 years.

## 7. Types of Inorganic NPs Utilized for Active Breast Cancer Targeting PDT Treatments

Inorganic NPs have great advantages over organic nanomaterials through their high stability, tunable size, and optical properties, as well as ease of surface functionalization to make them more biocompatible within biological applications [27,28]. Additionally, metallic and inorganic NPs have a lower degradation rate when compared to organic NPs [29]. The main characteristics of inorganic NPs have been summarized in Table 1.

### 7.1. Noble Metal Nanoparticles

Among various types of metallic NPs, gold NPs are ideal candidates for PS delivery into the body, due to their inertness, low toxicity, and limited side effects, as well as ease of synthesis and surface functionalization [30]. Furthermore, gold NPs are able to enhance the passive uptake of a PS-carrier system in tumor cells via the EPR effect [30,31]. Moreover, gold NPs possess a large surface area, which can be functionalized with a variety of ligands for active targeting [32]. Considering the high binding affinity of gold to thiol and amine groups, these NPs can be easily functionalized with antibodies, proteins, nucleic acids, and carbohydrates, which enable selective targeting and enhanced PS delivery in cancer tissues [33]. pharmaceutics-13-00296-t001_Table 1Table 1Main properties and structures of inorganic nanoparticles (NPs).Inorganic NPsPropertiesStructureReferenceGold NPsHigh surface to volume ratio, easy functionalization with antibodies, suitable for passive and active targeting, near infrared absorption, localized surface plasmon resonance (LSPR) characteristics
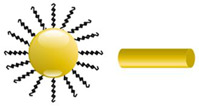
[34]Magnetic NPsSelective destruction of cancer cells due to heat release, superparamagnetism, and high field irreversibility
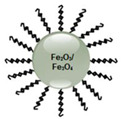
[35]Carbon-based NPsHigh strength, electron affinity, water solubility, and biocompatibility
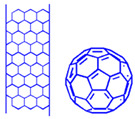
[36,37]Quantum dotsBroad excitation and narrow emission spectra, with high quantum yields and photostability
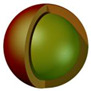
[38]Silica NPsHigh biocompatibility and stability, with easy surface functionalization
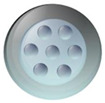
[28]Upconversion NPsUsed for the treatment of deep-seated tumors and exhibit lower phototoxicity 
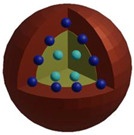
[28,39]Ceramic NPsControlled release of drugs, easy incorporation of hydrophilic and hydrophobic drugs, with high loading capacity
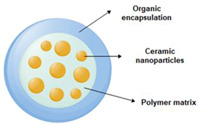
[40]


Gold NPs can be employed for imaging contrast agents and radiosensitizers thanks to the high atomic number of gold [30]. Furthermore, since gold NPs have a high atomic number and optical properties of light absorbance within near infrared (NIR) wavelengths [14], they generate heat when exposed to NIR laser irradiation, through surface plasmon resonance effects, allowing them to induce hyperthermia in tumor cells and so assist in improving PDT treatment outcomes [30]. In addition, since gold NPs peak absorbance wavelength is within the visible range of 400–600 nm, NIR light is transmitted through normal cells with low absorption [41], resulting in hyperthermia induction in cancer cells, with very little damage to surrounding healthy cells [42]. Lastly, the surface plasmon resonance effect of gold NPs within the NIR region enhances singlet oxygen and ROS generation [28], and so they tend to improve the overall treatment effect of PDT [31].

Studies by Li et al. (2009) noted that the passive tumor uptake of gold PS nanoconjugates in BC cells could be enhanced by binding them to active targeting biomarkers [43]. With respect to active targeting, a novel 4-component anti HER-2 antibody–zinc–phthalocyanine derivative–polyethylene glycol–gold NP conjugates were prepared for the in vitro PDT treatment of SK-BR-3 (BC cells with HER-2 receptors), MDA-MB-231 (BC cells without the receptor overexpression), and normal breast cells (MCF-10A) [44]. The study noted that the binding of the antibody to the gold PS nanoconjugate did not have an effect on the rate of singlet oxygen production and fluorescence microscopy demonstrated higher BC cellular uptake in SK-BR-3 cells, due to active HER-2 receptor targeting [44]. Within PDT treatments using 633 nm laser irradiation, the gold PS antibody nanoconjugate induced 40% cell death in SK-BR-3 cells, whereas MDA-MB-231 only noted 25%, and normal MCF-10A reported 7% cell death [44]. These findings suggested that active antibody receptor targeting enhanced the delivery of the PS in BC, which has over expressed HER-2 receptors, and so significantly improved the overall treatment outcomes of PDT [44].

In a similar study, gold NPs were stabilized with hydrophobic zinc phthalocyanine PS (C11Pc) and hydrophilic polyethylene glycol (PEG) for the PDT treatment of SK-BR-3 BC [45]. The C11Pc-PEG gold NPs were then further functionalized with jacalin (a lectin specific for cancer-associated Thomsen–Friedenreich (T) carbohydrate antigen) or with monoclonal antibodies specific for the human epidermal growth factor receptor-2 (HER-2) [45]. The study revealed that the NP conjugates were more specifically internalized within the acidic organelles SK-BR-3 BC cells [45]. Within PDT treatments under 633 nm irradiation, both jacalin and antibody conjugates at C11Pc equivalent concentrations of 1 μM and 1.15 μM, showed 99% cell death [45]. However, antibody-conjugates note the main advantages of limited PS dark toxicity, when compared to the jacalin-conjugates in SK-BR-3 BC cells, since prior to irradiation, antibody-conjugates reported a 58.9–70.2% viability, whereas jacalin-conjugates noted 85.5–98.5% [45].

Relating to the PDT effect of zinc phthalocyanine PSs on BC cells, gold NPs were functionalized with two substituted zinc (II) phthalocyanine PSs, with differing carbon chain lengths (C3Pc or C11Pc), a lactose derivative for stabilization, and a BC galectin-1 targeting agent [32]. Theses functionalized NPs–PSs were utilized to evaluate in vitro PDT efficiency of two breast adenocarcinoma cell lines namely, SK-BR-3 and MDA-MB-231 [32]. The conducted studies showed that the galectin-1 receptors overexpressed on the surface of MDA-MB-231 cells could only be targeted via the lactose-C3Pc-AuNPs, whereas the lactose-C11Pc-AuNPs in SK-BR-3 cells reported no active galectin-1 targeting. Furthermore, post-PDT (at 633 nm) no internalization and cell death was observed in MDA-MB-231 cells treated with lactose-C11Pc-AuNPs. Whereas, the lactose-C3Pc-AuNPs reported significant galectin-1 receptor targeting in both BC cell lines and noted far higher cytotoxicity in comparison to the C11Pc PS [32].

The PDT effect of gold NPs prepared via biphasic and monophasic approaches on SK-BR-3 in vitro cultured human BC cells was further elaborated by Penon et al. (2017) [46]. The gold NPs were further functionalized with a porphyrin derivative and PEG (PR-AuNP-PEG) synthesized using two different protocols [46]. The monophasic method reported more porphyrin derivative attached ligands per NP and higher singlet oxygen species yields than when compared to the biphasic nanoconjugates [46]. The researchers then covalently linked an anti-erbB2 antibody (PR-AuNP-PEG-Ab) to the monophasic PR-AuNP-PEG nanoconjugates, to target the overexpressed erbB2 receptors on the surface of SK-BR-3 BC cells [46]. Overall, the study noted higher cellular PR PS targeted uptake in BC cells when compared to normal cells, suggesting it had solubilized the PS, with significant cellular damage after 495 nm laser irradiation PDT treatment [46].

In other studies, gold nanostars have showed promising characteristics for Raman diagnostics. Since gold nanostars have tunable plasmon bands in the NIR tissue optical window, as well as multiple sharp branches, these act as “hot-spots” and so are capable of the surface-enhanced Raman scattering (SERS) effect [47]. Inspired by the unique SERS properties of gold nanostars, Fales et al. (2013) proposed the utilization of these nanostars for the Raman imaging and PDT treatment of BT-549 BC cells [48]. The nanotheranostic system comprised of a Raman-labelled gold nanostar, protoporphyrin IX (PpIX) PS, and a cell-penetrating peptide (CPP) known as transactivator of transcription (TAT) to enhance PS intercellular accumulation of the nanoplatform [48]. The gold nanostars were also coated with PEG and silica shells to enhance particle stability and PS-loading capacity [48]. Raman imaging results noted that the nanoplatform actively accumulated in the BC cells, due to the overexpression presence of TAT peptides [47]. The PDT treatment of BT-549 BC cells with 0.1 nM nanoconjugate and 633 nm irradiation revealed a higher photocytotoxicity and cell death, when compared to the 0.1 nM PpIX-loaded NP platforms without TAT [48].

The application of gold nanomaterials has been further developed to gold nanorod applications for successful BC active targeting PDT treatments. The enhanced active uptake of gold nanorods within in vitro cultured MCF-7 BC cells was found when gold nanorods were conjugated with anti-HER-2 antibodies [49]. Dube et al. (2018) reported that the conjugation of a complex of glycosylated zinc phthalocyanine to gold nanorods (AuNRs) could improve triplet, singlet, and fluorescence quantum yields, more than gold nanospheres (AuNSs), in PDT applications [50]. PDT results at 680 nm noted that less than 50% viable MCF-7cells were found at a concentration of ≥40 μg/mL complex-AuNRs, while this same result was only achievable at a concentration of ≥80 μg/mL complex-AuNSs, suggesting that AuNRs improve PS uptake and PDT outcomes at far lower concentrations [50].

A nanoplatform of AuNR@MSN-RLA/CS(DMA)-PEG was also proposed for the combinational PDT/photothermal therapy (PTT) of MCF-7 breast cancer [51]. Gold nanorods were coated with mesoporous silica (AuNR@MSN) and β-cyclodextrin (β-CD), as well as loaded with Indocyanine green (ICG) [51]. The nanoplatform was then grafted with an Ada modified RLA peptide ([RLARLAR]_2_), to enhance plasma membrane permeability and mitochondria-targeting capacity to form AuNR@MSN-ICG- β-CD/Ada-RLA [51]. Then 2,3-dimethylmaleic anhydride (DMA)-modified chitosan oligosaccharide-block-poly (ethylene glycol) polymer (CS(DMA)-PEG) was coated onto the surface of AuNR@MSN-ICG- β-CD/Ada-RLA to serve as a charge-switchable and anti-fouling layer (Figure 3) [51]. Within in vitro MCF-7 BC cytotoxicity assays, the nanoconjugate showed no obvious toxicity prior to laser irradiation [51]. MCF-7 cells treated with AuNR@MSN-ICG- β-CD/Ada-RLA/CS(DMA)-PEG at pH 6.8 displayed a higher inhibition and cellular uptake when compared to AuNR@MSN-ICG, suggesting that CS(DMA)-PEG coating protected the nanoplatform from hydrolysis, and so promoted cancer cell uptake [51]. Moreover, the weak acidity microenvironment of cancer cells could reverse the charge of the AuNR@MSN-ICG- β-CD/Ada-RLA/CS(DMA)-PEG nanoplatform, promoting mitochondrial targeting and overall improved ROS generation [51]. When MCF-7 BC cells were treated with the nanoconjugate and 808 nm NIR laser irradiation, the AuNR@MSN-ICG- β-CD/Ada-RLA/CS(DMA)-PEG complex noted the highest PDT and PTT inhibition relative to control groups, due to its overall stability and superior ROS, which was mediated by the plasmonic photothermal effects and local electric field of the DMA AuNR [51]. In vivo investigations within xenograft nude mouse models noted a higher tumor temperature in mice treated with AuNR@MSN-ICG- β-CD/Ada-RLA/CS(DMA)-PEG than when compared to those treated with AuNR@MSN-ICG, suggesting the DMA coated nanoconjugates were far more superior at treating BS in combination with PDT and PTT therapy [51].

A chlorine e6 PS based (Ce6)-AuNR@SiO_2_-d-CPP nanoconjugate template was developed by synthesizing gold nanorods and passivating PEGylated mesoporous SiO_2_ onto the gold NPs surface core to entrap the Ce6 PS [52]. Then, a D-type cell penetrating peptide (d-CPP) was linked to the gold nano shell to direct active PS targeting of the nanocarrier towards human BC MCF-7 cells [52]. Free Ce6, AuNR@SiO_2_-mPEG, and Ce6-AuNR@SiO_2_-d-CPP showed no dark cytotoxicity within in vitro cultured BC cells. The combinative PDT (650 nm)/PTT (808 nm) therapy results on BC cultured cells noted that Ce6-AuNR@SiO_2_-d-CPP provided the highest treatment outcomes and caused almost complete cell death [52]. The injection of Ce6-AuNR@SiO_2_-d-CPP into a nude mouse BC xenografts and exposure to PTT/PDT combinational therapy noted a significant decrease in tumor weight sizes [52].

Dendrimer-encapsulated NPs (DENs) were first reported in 1998, whereby metal ions were encapsulated within dendrimers, and to reduce them to produce zerovalent DENs [37]. DENs are synthetic polar macromolecules consisting of branches that emanate from a core that has functional groups of neutral, positive, or negative charges [53]. They are monodisperse NPs, which have lower toxicity in cells, a high surface reactivity, as well as allow for slow release and, thus, report a great accumulation in tumor cells, making them suitable for various drug delivery enhancement applications [54]. Poly(propyleneimine) (PPI) and poly(amidoamine) (PAMAM) dendrimers are some examples of DENs [55].

In this regard, one study evaluated the applicability of multiple particles delivery complexes (MPDC) for the PDT treatment of MCF-7 BC cell, using 680 nm laser irradiation [56]. The MPDC was comprised of gold NP encapsulated dendrimers (AuDENPs) and a sulfonated zinc-phthalocyanine mix (ZnPcs_mix_) [56]. The morphology of the PDT AuDENPs–ZnPcs_mix_ treated BC cells noted an altered appearance from epithelial-like to irregular and a 59% of apoptotic cell death was found, in comparison to control groups [56]. In addition, a decrease in the polarized mitochondrial membranes of the BC cells and an increase in the depolarized cell membranes were observed after PDT treatment, with an increase in caspase 3/7 activity and cytotoxicity being found [56].

Recently, a multi-stimuli-responsive theranostic nanoplatform for the fluorescence imaging-guided PDT/PTT dual-therapy of MCF-7 BC cells was proposed [57]. The nanoplatform was based on functionalizing AuNRs with hyaluronic acid (HA), and subsequently conjugating anti-HER-2 antibody, 5-aminolevulinic acid (ALA) and Cy7.5 onto the HA, to enhance active PDT targeting and fluorescence imaging respectively (Figure 4) [57]. Cellular uptake efficiency of AuNR-HA^-ALA/Cy7.5^-HER-2 noted a significantly improved uptake of 75.5% in MCF-7 cells when compared to control groups, which received AuNR-HA^-ALA/Cy7.5^ of 36%, suggesting the nanoplatform improved PS uptake via the specific HER-2 receptor mediated dual-targeting strategy [57]. Furthermore, AuNR-HA^-ALA/Cy7.5^-HER-2 single treated PDT MCF-7 cells at 635 nm reported a 75.6% decrease in cell viability, and cells treated with singular PTT at 808 nm noted a 58.4% decrease in cell viability [57]. Overall, a combinative PDT/PTT modality at a 5.5 μg/mL ALA concentration with the AuNR-HA^-ALA/Cy7.5^-HER-2 nanoplatform noted a significant 61.2% cell death [57]. Within in vivo studies on BC-induced mice, this dual treatment modality showed a rapid decrease in tumor sizes 20-days post treatment [57].

NP drug carriers are sometimes easily recognized and cleared from the body via the mononuclear-phagocyte system (MPS); thus, the surface of NPs drug carriers are generally coated with PEG to act as a shield, and so reduce this biological clearance [58]. However, some studies have noted that upon second administration of PEGylated NP drug carriers the human body can sometimes become stimulated to produce anti-PEG antibodies, resulting in the unwanted rapid clearance of the PEGylated NP, decreasing overall drug uptake in tumor cells [58,59,60]. In order to alleviate the rapid clearance of NPs via MPS, researchers have focused on red blood cells (RBCs), since they are a natural long-circulation delivery vehicle, which do not interfere or impact on the functionality of NPs [61,62,63].

In a study, cationized gold nanoclusters (CAuNCs) with various initial sizes of 150, 200, and 300 nm were constructed and coated with HA (CAuNCs@HA) [58]. In order to increase the circulation of the CAuNCs@HA nanoclusters, an RBC membrane was attached to its surface forming mCAuNCs@HA [58]. The mCAuNCs@HA nanoclusters were then loaded with a pheophorbide A (PheoA) PS, which is a ROS-responsive prodrug paclitaxel dimer (PXTK) and an anti-PD-L1 peptide dPPA forming pPP-mCAuNCs@HA [58]. This combinative PDT, chemotherapy and immunotherapy treatment approach was investigated within in vitro cultured 4T1 BC cells [58]. The study confirmed that the RBC membrane improved the overall PS in vitro cellular uptake in 4T1 cells, which was found to be a 2.02-, 1.55-, and 1.95-fold higher uptake for NP-300, NP-200, and NP-150 nm pPP-mCAuNCs@HA, respectively, when compared to uncoated groups [58]. The study also demonstrated that the 650 nm laser irradiation PDT induced late apoptosis with pPP-mCAuNCs@HA was a 2.41-fold higher than when compared to pPP-mCAuNCs@HA without irradiation [58]. The anti-tumor therapeutic effects within in vivo 4T1 tumor bearing female mice treated with pPP-mCAuNCs@HA and irradiation was 2.47-fold higher than when compared to groups injected with pPPmCAuNCs@HA without irradiation [58].

The successful applications of gold NPs in BC PDT active targeting treatments within in vitro and in vivo studies have led to more clinical applications [64]. In spite of the fact that gold NPs are inert for bio-tissues and are an alternative platform for PS delivery in PDT BC treatment studies, particular care must be exercised within clinical studies to noted their long-term toxicity and biodistribution, as some tend to have a limited clearance in the spleen and liver; however, this phenomenon is highly dependent on the different shape, size, and surface charge of AuNP [64].

### 7.2. Magnetic Nanoparticles

Magnetic NPs (MNPs) have drawn tremendous attention within in vivo and in vitro biomedical uses, because of their high field irreversibility, small size, and surface functionality [35,65]. Within in vitro studies, MNPs have been employed in magnetorelaxometry, diagnostic separation, and selection applications, whilst within therapeutic studies they have been utilized to induce hyperthermia and promote active drug-targeting, as well as assist in diagnostic applications, such as nuclear magnetic resonance imaging (NMR) [66].

Narsireddy et al. (2014) fabricated chitosan coated Fe_3_O_4_ NPs, which were deposited with gold NPs followed by lipoic acid conjugation [67]. A 5,10,15,20-tetrakis(4-hydroxyphenyl)-21H,23H-porphyrin PS was also attached onto the surface of gold NPs to form Fe_3_O_4_-Au-LA-PS (MGPS) [67]. In order to improve targeting of this nanoconjugate, human epidermal growth factor receptor specific peptide (Affibody HER-2) was anchored onto its chitosan coat forming Aff-MGPS [67]. The PDT effects of Aff-MGPS were then investigate within in vitro cultured SK-OV-3 BC cells, which are HER-2 positive [67]. The cellular uptake efficiency of the targeted Aff-MGPS was far more superior than when compared to free PS or NPs controls alone (Figure 5) [67]. In addition, no dark toxicity was observed for MGPS or Aff-MGPS nanoplatforms, while free PS noted high dark toxicity [67]. Furthermore, the Aff-MGPS nanoplatforms noted improved targeted peptide uptake in BC cells when compared to group controls [67]. The targeted PDT specific delivery of Aff-MGPS was further assessed in nude SK-OV-3 BC induced tumor mice, in comparison to MGPS treatment alone at 120 J/cm^2^, and tumor volumes in mice grew slower in Aff-MGPS PDT treated mice than when compared to MGPS irradiation treatment alone, suggesting Affibody HER-2 enhanced BC PDT targeted treatment outcomes [67].

The photodynamic anticancer activities of magnetic Fe_3_O_4_ NPs on BC was further investigated through the conjugation of Ce6 and FA onto its surface to form Fe_3_O_4_-Ce6-FA [68]. The synthesized nanoconjugate could effectively produce ^1^O_2_ and ROS, with no dark toxicity being found [68]. Within PDT Fe_3_O_4_-Ce6-FA 660 nm experiments, MCF-7 in vitro BC cells reported a concentration-dependent manner decrease in viability, and increased apoptotic cell death pathway activation via caspase 3/7, with notable nuclear fragmentation and plasma membrane translocation [68].

Within a study conducted by Matlou et al. (2018) the aim was to assess the PDT activity of two zinc phthalocyanine (Pc) derivatives: Zn mono cinnamic acid phthalocyanine and zinc mono carboxyphenoxy phthalocyanine complexes, which were covalently linked to a FA targeting agent and an amino functionalized Fe_2_O_3_ MNP (AMNPs) [69]. The dark toxicity of this MNP PS carrier noted a significant decrease after attachment of the FA complex [69]. The in vitro MCF-7 PDT effect of Pc-AMNPs noted a significant 60% cell death under 670 nm irradiation, when compared to Pc-FA, which only reported a 40% cell death [69].

Fe_3_O_4_ and Fe_2_O_3_ NPs are supermagnetic [70], and so release a significant amount of heat upon external exposure to laser irradiation and, thus, these NPs are highly effective in combinative PTT and PDT applications to destroy cancer cells [71]. Furthermore, when these magnetic NPs are PEGylating or bound to other polymers, their rapid clearance from the MPS can be alleviated [72]. However, PDT applications with MNPs cannot be taken lightly, as free Fe^2+^ may react with oxygen or hydrogen peroxide to form Fe^3+^ and hydroxyl radicals, which are toxic and can damage DNA; thus, the confirmation of the stability of these NP as PS carriers is crucial [70].

Even though only a few studies have been performed using MNPs for the active PDT treatment of BC, their superparamagnetic PTT hyperthermia properties have been demonstrated as a powerful and efficient approach in clinical trials of unresectable tumors or cancers representing terminal illness [73]. Furthermore, MNPs can be utilized in multiple therapeutic and diagnostic strategies [73], as well as eradicating apoptosis resistant cancer cells, since they generate heat intracellularly within the lysosomes and the tumor stroma, and so can obliterate tumor cells completely via necrotic cell death [74,75]. Thus, further studies and investigations utilizing MNPs for the effective treatment of BC is an ongoing need.

### 7.3. Carbon-Based Nanoparticles

Fullerene, carbon nanotubes, and graphene are carbon nanomaterials, which are commonly utilized as PS nanocarriers in PDT applications [76,77,78]. When PSs are attached via covalent or non-covalent bonding to functionalized carbon-based nanomaterials, they often provide improved solubility and biocompatibility in PDT treatments [36]. Fullerenes are carbon-based nanomaterials, which present in the forms of tubes, ellipsoids, or spheres, and successfully produce ROS upon irradiation exposure at an appropriate wavelength [28]. Single walled or multi-walled carbon nanotubes are other types of PS nanocarriers used in PDT cancer treatments [28]. Since carbon nanotubes present advantageous characteristics, such as fast elimination, low cytotoxicity, ease of functionalization, and reliable internalization through endocytosis, it makes them ideal PS nanocarriers in PDT [28]. Regarding graphene nanomaterials, due to their large surface areas they offer high therapeutic loading capacities for enhanced PS uptake in tumor cells [79].

In studies performed by Shi et al. (2014), fullerene-iron oxide NPs (IONP) were synthesized and functionalized with PEG, Ce6, and FA (C60-IONP-PEG-FA) for the active tumor targeting [80]. The performance of these multifunctional NPs was studied for their PDT effect, radiofrequency thermal therapy (RTT), and magnetic targeting, within in vitro MCF-7 BC cells and in vivo BC mice models [80]. Individual in vitro PDT assays at concentration of 16 µg/mL C60-IONP-PEG-FA and 532 nm laser irradiation, 31.3% viability was reported, and individual RTT therapy at the same concentration with 13.56 MHz radiofrequency noted a 36.9% of viable cells [80]. In combinational C60-IONP-PEG-FA RTT, followed by PDT, in vitro assays reported a significant 18.8% of cells only being found to be viable [80]. Within in vivo studies on S180 BC tumor-bearing mice, individual PDT and PTT applications induced 62% and 37% of apoptosis, respectively, and the integration both treatments could enhance apoptotic cell death to 96% [80].

By taking advantage of the ultra-high loading capacity of graphene oxide (GO) through π−π stacking and hydrophobic interactions [81], GO(HPPH)-PEG-HK was prepared by functionalizing it with PEG-GO, so that a HK peptide (which binds specifically to integrin αvβ6 on BC tumors) could be linked to it [82]. This actively functionalized nanoconjugate was then coated with a Photochlor (2-[1-hexyloxyethyl]-2-devinyl pyropheophorbide-alpha, HPPH) PS [83]. The large surface area of GO enabled conjugation of multiple HK peptides and high concentrations of HPPH, allowing for improved in vivo PDT treatment outcomes of BC [83]. Within in vivo 4T1 BC tumor mouse models, results reported that GO(HPPH)-PEG-HK under 671 nm laser irradiation that tumor growths remarkably decreased in comparison to control groups [83]. Furthermore, studies went on to note that remarkable increases in the CD40^+^ and CD70^+^ fractions in treated mice after PDT treatment with GO(HPPH)-PEG-HK induced dendritic cell maturation and so promoted anti-tumor immunity in the 4T1 tumor model (Figure 6) [83]. Thus, this actively functionalized nanoconjugate via PDT application could reduce in vivo BC tumor sizes, as well as trigger host anti-tumor immunity, to cause the inhibition of metastasis and further tumor growth [83].

Overall, very few studies have been conducted using carbon-based nanomaterials for the active PDT targeting of BC. However, from the above studies, it can be seen that the combination of the synergistic effects of carbon–based NPs with PSs can improve the efficacy of BC PDT treatment [76]. It is worth mentioning that fullerene cages, such as C60 [84] and carbon nanotubes [85], can act alone as PSs in PDT applications and so self-generate ROS from photons owing to their π bond electrons [86]. Thus, fullerene derivatives alone are competitive PSs for in vivo PDT or preclinical treatment [87], since no additional PS is required to generate ROS and, thus, should be researched further for BC treatment [76]. Despite all of the idealistic properties carbon nanotubes have, they sometimes can induce asbestos-like inflammation [73], which is a carcinogenic, and so their individual toxicity needs to be fully investigated and understood [88].

### 7.4. Semiconductor Quantum Dots (QDs)

Quantum dots are a subclass of fluorescent nanomaterials, which have unique chemical and physical properties compared to organic dyes [89]. They have been utilized as multifunctional nanocarriers for PDT thanks to their high quantum yields, simple surface modification, and tunable optical properties [90]. They are also excellent donors in fluorescence resonance energy transfer (FRET) applications [91]. To date, no studies have been investigated relating to the application of active targeted QDs for the PDT treatment of BC.

Studies by Monroe et al. (2019) only performed a spectrophotometric assay in order to assess the cellular uptake, cytotoxicity, and ROS generation of graphene QDs (GQDs) associated with methylene blue (MB) PS against in vitro cultured MCF-7 BC cells [92]. This study reported that MB improved cytotoxicity and ROS generation when compared to a 1:1 GQD:MB ratio [92].

In another study, Zn(II) phthalocyanines (ZnPcs) with different substitutes were fabricated and conjugated to GQDs to investigate the in vitro PDT activity of Pc-GQDs conjugates in a human BC in vitro MCF-7 cell line [93]. The conjugate and Pcs alone did not report dark toxicity and in vitro PDT studies noted that Pc-GQDs conjugates enhanced treatment outcomes when compared to Pcs administration alone [93].

The above QD-based BC PDT studies have paved a new avenue for researchers to synthesize various targeted-QDs against breast cancer. It has been reported that QDs have potential cytotoxicity under UV irradiation to act as efficient PSs [90]. Furthermore, NIR fluorescent QDs provide improved PS water solubility, chemical stability, and low optical interference with biological tissues in PDT cancer treatments, when compared to small molecule-based PSs administered alone [94]. Additionally, the large surface area of QDs also enables the conjugation of multiple PSs and ligands for targeted photodynamic imaging [90]. Nevertheless, one of the most controversial problems with QD-based PDT is the high toxicity they possess, since most consist of toxic heavy metals, such as cadmium ions, and so they tend to be under investigated [90,95]. In order to alleviate some of these issues cadmium free QDs, such as zinc and indium based QDs [96], substituted with other elements, such as silicon or carbon [97], or their incorporation into polymeric NPs [98] have been proposed to enhance their application in diagnostic applications and PDT cancer clinical trials [99]. Overall, it is envisaged that QDs, particularly GQDs, will open the door to a multitude of new opportunities for PDT treatment of BC, and through continued research, they could subsequently be able to provide high biocompatibility and improved PS solubility in biological media, with less unwanted toxic effects.

### 7.5. Ceramic Nanoparticles

Ceramics NPs are inorganic solids made up oxides, carbides, carbonates, and phosphates, and have properties that range between metals and non-metals [40]. Silica (SiO_2_), titanium oxide (TiO_2_), alumina (Al_2_O_3_), zirconia (ZrO_2_), calcium carbonate (CaCO_3_), and hydroxyapatite (HA) are some examples of ceramic NPs with porous characteristics that can be fabricated to control the release of drugs [40]. The main features of ceramic NPs are high loading capacity, stability, and chemical inertness, as well as heat resistance and ease of conjugation to either hydrophilic or hydrophobic drugs [29], making them highly advantageous for drug delivery, imaging, photodegradation of dyes, and photocatalysis applications [37,40,100].

#### 7.5.1. Silica Nanoparticles

Silica is an oxide of silicon and one of the most efficient materials for controlled drug delivery, since it can store and be controlled to gradually release therapeutic drugs [40]. Mesoporous silica nanoparticles (MSNs) are formed by polymerizing silica, and so have distinctive properties, such as tunable pore size (allowing for a high therapeutic drug loading capacity capacities) [101], large surface area to volumes, automatic release of drugs, as well as ease of functionalization with various functional groups or ligands, offering actively targeted drug delivery capabilities [40]. Thus, within PDT applications, PSs can be easily covalently linked or encapsulated onto the surface of silica NPs for favorable cancer treatment outcomes [28].

A nanosystem comprised of mesoporous silica NPs (MSN) with covalent anchoring of a synthesized anionic porphyrin PS and BC targeting mannose was presented by Brevet et al. (2009) [102]. The study confirmed that PS mannose-functionalized NPs within in vitro cultured MDA-MB-231 BC cells improved the efficiency of PDT relative to the non-functionalized NPs, since it induced 99% cell death when irradiated at 630–680 nm with 6 mW/cm^2^, while non irradiated control groups only noted 19% cell death [102].

Another promising nanoconjugate PS depended on a two-photon absorption, which integrated a two-photon excitation (TPE) with silica nanotechnology [103,104]. Conventional PSs require the absorption of a single photon equal to the band-gap energy of a PSs [105]. However, when a PS absorbs two lower energy photons of infrared light, TPE can occur and the sum of the photon energies are equal to the band-gap of energy, leading to a deeper light penetration and lower photo-bleaching of the actual PS [105,106,107,108,109]. Furthermore, in a TPE, the nonlinearity of photon absorption allows a PS activation to occur at the focal point of a laser beam, and so allows for greater spatial control of PS activation in three-dimensional (3D) tumor models, decreasing off-target phototoxicity in surrounding healthy tissues [110,111].

With respect to the TPE–PDT, a porphyrin functionalized porous silica NP (pSiNP) was coupled to a mannose targeting moiety to investigate the imaging and PDT potentials within in vitro cultured MCF-7 BC cells [112]. When compared to other two-photon absorbing nanoparticles such as, CdSe quantum dots, gold nanorods, or carbon dots, pSiNPs appear to be biodegradable in vivo [113], since their silicon components degrade to silicic acid, which can quickly be eliminated by kidneys [112,113]. The authors of this showed that the pSiNP with mannose moieties were able to actively accumulate in MCF-7 BC cells, with far higher PDT efficacy, since phototoxicity results noted a 2.3-fold better outcome for two photon PDT at 800 nm than when compared to one-photon excitation at 650 nm [112].

Within studies performed by Cao et al. (2014), in order to enhance PS accumulation in BC cells, it was proposed to use mesenchymal stem cells (MSCs) to directly deliver a PS to in vitro cultured MCF-7 cells [114]. The application of MSCs in PDT cancer treatments seems promising, as various studies have demonstrated that they have a naturally high tumor affinity within in vivo tumors, they can be easily isolated from bone marrow and modified to carry desired drugs, as well as be efficiently implanted into patients to avoid immune system clearance [115,116,117,118]. In this study, porous hollow silica NPs were conjugated to a purpurin-18 PS (PS-SiO_2_NPs) and then they were loaded into the MSCs cells (PS-SiO_2_NPs-MSCs) for in vivo PDT studies in MCF-7 modified mouse models (Figure 7a) [114]. Results noted that the BC tumor affinity of the MSCs was not inhibited by loading PS-SiO_2_NPs into the MSCs, and that intercellular ROS generation proportionally increased with PS-SiO_2_NPs-MSCs conjugation upon laser irradiation, suggesting the in vivo BC tumors retained the PS [114]. In addition, within PDT studies, the in vivo groups that received PS-SiO_2_ NPs-MSCs reported far greater tumor growth inhibition than when compared to control groups, which received unmodified MSCs without loading PS-SiO_2_NPs, suggesting that the MSCs cells were capable of high PS BC tumor affinity targeting (Figure 7b) [114].

Studies by Bharathiraja et al. (2017) reported that silica NPs decorated with Ce6 and FA (silica-Ce6-FA) could accumulate far higher within in vitro cultured MDA-MB-231 BC cells, when compared to free Ce6 [119]. Even though the level of ROS generated by silica-Ce6-FA nanoconjugate was moderately lower than when compared to free Ce6, at 680 nm PDT, the study showed that due to the folate receptor targeting in the nanoconjugate, the PS uptake in BC cells was improved, and so higher cell death was observed than when compared to free Ce6 administration alone [119].

Although, NIR light within the range of 630–800 nm is employed as an excitation PDT source to treat deep-seated cancer tissues [120], most clinically approved PSs have a low absorption in NIR region, and so their overall penetration depth is less than 1 cm [121]. Thus, researchers have also begun to investigate X-ray sources for use in X-ray-mediated PDT (X-PDT), since they are able to penetrate far deeper into tissues, which perhaps better outcomes [112]. Within these applications PSs need a system to convert X-rays into UV–visible photons, since they cannot absorb X-ray photons directly [112]. Scintillating nanoparticles or nanoscintillators, such as lanthanide doped rare-earth nanoparticles [122], have emerged as energy transducer for this conversion and deep seated X-PDT treatment [123,124,125].

Studies by Sengar et al. (2018) investigated X-PDT for the deep penetration of BC tumors [126]. They synthesized Y_2.99_Pr_0.01_Al_5_O_12_-based (YP) mesoporous silica (MS) coated NPs and functionalized them with PpIX and FA (YPMS@PpIX@FA) for the X-PDT treatment of BC cells with overexpressed folate receptors (*Folr 1*) [126]. The utilized BC in vitro cell lines were PyMT-R221A mouse BC cells (which have high levels of folate receptors), as well as 4T1 BC cells (which have low folate expression) [126]. PyMT-R221A mouse BC cells reported higher cellular uptake of YPMS@PpIX@FA when compared to 4T1 BC cells, revealing that FA targeting of this nanoconjugate was functional [126]. Additionally, non-activated YPMS@PpIX@FA reported a low cytotoxicity when used at concentrations below 25 µg/ml, while upon light activation at 365 nm, a remarkable decrease in PyMT-R221A mouse BC cells was observed [126]. Lastly, administration of YPMS@PpIX@FA suspension at a single dose of up to 125 mg/kg did not cause the death or any detectable behavior in inoculated CD1 mice [126].

Overall, it can be observed from the above in vitro and in vivo studies that silica NPs seem very promising for the PDT treatment of BC, and due to their non-toxicity and rapid renal clearance, the move forward of these studies into clinical applications is pertinent.

#### 7.5.2. Titanium Oxide Nanoparticles

Titanium dioxide (TiO_2_), also called titania, is another type of ceramic NP, which possesses chemical and biological inertness, photostability, photoactivity, and high stability within biomedical applications [127]. More importantly, the strong oxidizing and reducing ability TiO_2_ has when photoexcited with irradiation at <390 nm can produce ROS, which consequently induces apoptotic cell death in BC cells [128].

Studies by Gangopadhyay et al. (2015), constructed TiO_2_ NPs and decorated them with a 7,8-dihydroxy coumarin PS chromophore and chlorambucil (Ti-DBMC-Cmbl NPs) and FA (Ti-FA-DBMC-Cmbl NPs) to serve as a chemotherapeutic drug and phototrigger, respectfully in PDT/chemotherapy treatments of in vitro cultured MDA-MB BC cells (Figure 8) [129]. After 60 min of PDT laser irradiation at ≥410 nm, cells treated with Ti-DBMC-Cmbl NPs noted a 35% cell viability, whereas cell treated with Ti-FA-DBMC-Cmbl NPs reported a mere 19% cell viability and more significant apoptotic cell death [129]. Overall, results revealed that the synergic effect of both targeted PDT and well known chemotherapeutic drug chlorambucil was successful for the eradication of MDA-MB BC cells [129].

In general, ceramic NPs show great potential in carrying PSs in PDT application to targeted BC tumors, due to their excellent chemical inertness and high heat resistance. However, since limited studies have been performed using ceramic NPs in BC, it is constructive to highlight that a suitable method in relation to their synthesis to control size, porosity, surface area to volume ratio, should be fine-tuned in order to allow for a high PS pay loading capacity and reduce any possible unwanted biological clearance issues [129].

### 7.6. Other Inorganic Nanoparticles

Cerium oxide NPs (or nanoceria/ceria NPs) are considered from a lanthanide metal oxide that can be used as an ultraviolet absorber [130]. They have antioxidant properties at a physiological pH, while their oxidases activity in tumors idealistically functions in an acidic microenvironment [131,132]. Nanoceria NPs alone tend to have a poor water solubility so they are generally coated with polymers to enhance biocompatibility, stability, and their overall solubility [133].

A multifunctional drug delivery system of PPCNPs-Ce6/FA was introduced by Li et al. (2016) and it comprised of Ce6/FA-loaded branched polyethylenimine–PEGylation ceria NPs (PPCNPs) for the possible PDT targeting of drug resistant in vitro BC MCF-7/ADR cells in combination with chemotherapeutic agent DOX [134]. The results revealed that internalization efficiency and diffusion of the synthesized nanoplatform with positive surface charges via endocytosis in BC cells was far higher than free Ce6 [134]. PDT efficiency under 660 nm irradiation in BC cells treated with the PPCNPs-Ce6/FA reported a 35% apoptotic and necrotic cell death, while BC cells treated with PPCNPs-Ce6 only noted an overall 25% cell death [134]. Moreover, results reported that low-dose PPCNPs-Ce6/FA PDT remarkably improved the chemotoxicity of DOX in in vitro MCF-7/ADR BC cells in a dose-dependent manner [134]. In vivo PDT studies within MCF-7/ADR athymic nude mouse xenograft models showed a significant 96% reduction in the tumor volume when injected with PPCNPs-Ce6/FA, in comparison to a 25% reduction when PPCNPs-Ce6 was applied [134].

In relation to metastatic triple-negative BC (mTNBC), numerous studies have noted that treatments, such as radiation, cytotoxic chemotherapy, and surgical interventions are ineffective and so this has driven researchers to consider immunotherapy for the possible treatment of mTNBC [135,136]. Tumor immunotherapy relies on the fact that BC cells can be eradicated by host cytotoxic CD8^+^ T cells [137,138].

In this regard, Duan et al. (2016) introduced a promising strategy using a checkpoint blockade-based immunotherapy for the treatment of primary in vitro 4T1 BC tumors [139]. A non-toxic core–shell comprising of ZnP@pyro NPs was fabricated using Zn and pyrophosphate (ZnP) and a pyrolipid PS was incorporated into its core, for PDT applications, while a PD-L1 antibody was added for checkpoint blockade immunotherapy [139]. Results reported that the immunogenic ZnP@pyro NPs were non-toxic prior to light activation [139]. Within PDT studies, they successfully eliminated in vitro BC cells upon 670 nm irradiation, through apoptotic and necrotic cell death [139]. PDT in vivo investigations of the ZnP@pyro NPs on orthotopic 4T1 tumor-bearing mice demonstrated that this immunogenic PS nanocarrier enhanced PS uptake via the EPR effect for high tumor accumulation, as well as disrupted tumor vasculature and increased tumor immunogenicity [139]. The authors also claimed that the ZnP@pyro NPs not only prohibited the further metastasis, but also inhibited pre-existing metastatic tumors growth by generating systemic anti-tumor immunity [139].

### 7.7. Upconversion Nanoparticles

The unique “photon upconversion” process of upconversion NPs(UCNPs) has been applied in low tissue penetration depth PDT applications [140,141]. Upconversion is an anti-Stokes shift, which is defined as the conversion of NIR light to a shorter wavelength of light in the visible region [140]. Thus, UCNPs are able to absorb two or more low energy photons and, thus, show a unique anti-Stokes shift of fluorescence emission in UV–Vis wavelengths (300–700 nm) under NIR light excitation (750–1400 nm) [142]. UCNPs can be utilized in biomedical applications as they have shown improved reduced fluorescence background, with lowered phototoxicity [39]. Therefore, in order to treat deep-seated tumors, PSs in NPs are excited with longer wavelength [28] and emitted fluorescence by UCNPs, so can excite PS electrons effectively to produce efficient amounts of singlet oxygen in PDT applications [143].

In 1991, Cai et al., utilized novel TiO_2_ or ZnO semiconductors, which had photo-effects, such as inorganic PSs for the PDT treatment of cancer [144]. These semiconductors promoted electrons from valance bands to conduction bands upon PDT UV irradiation, leaving electron hole pairs [28] and so resulted in oxidation or reduction of chemical species, such as water and oxygen around the TiO_2_ or ZnO semiconductors, to generate ROS [145,146].

In this regard, Janus nanostructures comprised of NaYF4:Yb/Tm UCNPs with TiO_2_ inorganic PSs were synthesized by Zeng et al. (2015) and loaded with FA and DOX (FA–NPs–DOX) (Figure 9) for NIR-triggered inorganic targeted PDT and chemotherapy treatment of drug-sensitive MCF-7 and drug resistant MCF-7/ADR BC cells within in vitro and in vivo applications [147]. The chemotherapeutic results alone revealed that the FA-targeted nanocomposite promoted the cellular uptake of DOX, as well as caused a viability decline of 44.4% in MCF-7 and 28.9% in MCF-7/ADR BC cells [147]. However, combinational chemotherapy and PDT results under 980 nm NIR irradiation, noted a far higher significant decrease in cellular viability, whereby only 5.8% MCF-7 cells were found viable and 17.6% of MCF-7/ADR BC drug resistant cells were found to alive [147]. Within PDT in vivo assessments on female BALB/c (nu/nu) nude mice, MCF-7 tumor growths reported a 99.34% inhibition of growth, while MCF-7/ADR tumor growths noted a 96.74% decline, when treated with FA–NPs–DOX + NIR [147].

In studies by Zeng et al. (2015), HER-2-targeted multifunctional nanoprobes based on 808 nm-excitation bound to NaGdF4:Yb,Er@NaGdF4:Yb@NaGdF4:Yb,Nd UCNPs, with Ce6 PS and SiO_2_ (T-UCNPs@Ce6@mSiO_2_) were fabricated for 808 nm irradiation PDT and magnetic resonance imaging (MRI) within in vitro MDA-MB-435 BC cells [143]. Regarding the cellular uptake, the accumulation amount of T-UCNPs@Ce6@mSiO_2_ in in vitro BC cells was 1–2 times higher than those treated with UCNPs@Ce6@mSiO_2_, due to HER-2 active targeting [143]. Furthermore, the PDT treatment of in vitro BC cells using non-targeted UCNPs@Ce6@mSiO_2_ reported a 16.4% cell viability, whereas cells treated with T-UCNPs@Ce6@mSiO_2_ noted a 6.8% cell viability, suggesting that the PDT efficiency improvement was due to active targeting [143]. Additionally, in vitro T-UCNPs@Ce6@mSiO_2_ PDT, reported a significant 16.5% for early apoptosis and 10.2% for late apoptosis cell death [143]. The in vivo PDT investigation of nanocomposite in MDA-MB-435 tumor-bearing nude mice indicated that the targeted T-UCNPs@Ce6@mSiO_2_ significantly enhanced tumor accumulation of up to 12%, whereas the non-targeted UCNPs@Ce6@mSiO_2_ only noted a 2% accumulation potential [143]. Furthermore, the MR signal was far higher and stronger in T-UCNPs@Ce6@mSiO_2_ treated mice than when compared to those treated with UCNPs@Ce6@mSiO_2_ [143].

Within studies by Wang et al. (2017) Lanthanide-doped UCNPs were encapsulated in fourth-generation poly amido amine (PAMAM) dendrimers, bearing 64 peripheral amines (G4) and Ce6 to assess NIR-trigged PDT in 2D and 3D in vitro MCF-7 BC [142]. The internal cavities within the dendrimers enabled the trapping of small molecules through host-guest affinity and so enhance the cellular uptake of the UCNPs [142]. More importantly, when Ce6 was loaded onto the dendrimer-modified UCNPs, and 660 nm PDT laser irradiation applied in 2D models 50% cell death was noted, whereas approximately 70% cell death was found in those treated with 980 nm PDT laser irradiation [142]. These findings confirmed that the NIR could pass through the 2D BC tumor cell model membranes and organelles to reach and effectively activate the UCNPs, and so enhance PDT efficacy with deeper tissue penetration [142]. With reference to the 3D model, 980 nm NIR light noted a deep tumor penetration and produced cell death that was consistent with the 2D model results [142]. Moreover, in vivo PDT assessments using 980 nm irradiation in 4T1 BC tumor-bearing mouse models treated with high doses of the UCNPs, demonstrated significant tumor growth inhibition, through induction of the γH2AXser139 protein marker for DNA double strand breaks; thus, substantial DNA damage was observed [142].

A precise tumor-specific UCNP targeting strategy was assessed by Yu et al. (2018) for the enhanced PDT treatment of in vitro MCF-7 BC cells [148]. NaYF4:Yb^3+^, Er^3+^ UCNPs capped with polyacrylic acid (UCNPs@PAA) were fabricated and modified with FA and Ce6 PS, as well as were functionalized with DNA sequences of varying lengths (Figure 10A) [148]. The in vitro cellular uptake results showed that the fabricated UCNPs@PAA-DNA located efficiently within BC lysosomes via effective folate receptor targeting (Figure 10B) [148]. Significant reduction within in vitro 980 nm PDT treated BC cells with UCNPs@PAA-DNA was reported [148]. The study stated that the Ce6 PS on the longer DNA nanocomposite moved to the vicinity of the UCNPs and generated singlet oxygen upon NIR irradiation, through Förster resonance energy transfer (FRET). The study proposed that when the fabricated UCNP reached the BC tumor cells, the C base-rich long DNA within the nanocomposite could form a C-quadruplex and the FA groups overexpressed on the folate receptors of BC cells could be attracted, and so efficiently active BC tumor targeting was achieved [148]. Furthermore, the pre-protective strategy using UCNPs@PAA-DNA with longer DNA alleviated any other possible side effects on the normal cells, as the FA groups of the shorter DNA was protected by this longer DNA to preclude any possible binding with normal cell folate receptors [148]. In vivo PDT experiments within BALB/c mice with BC xenograft tumors not only demonstrated the successful accumulation of the nanocomposites, but also eliminated tumor volume tenfold in comparison to control groups [148].

Generally, NP particle size can affect their overall uptake and retention in the liver, kidney, and spleen [149], furthermore large sized NPs can induce high toxicity, with unwanted side effects and heightened cellular phagocytosis [150]. Thus, within clinical applications, they tend to be more inclined towards the use of small NPs, which report less retention and unwanted toxicity [151]. Within a study performed by Yu et al. (2018) a core–multishell nanocomposite (MNPs(MC540)/DSPE-PEG-NPY) was constructed that was an ultrasmall size for the in vitro and in vivo PDT evaluation within MCF-7 BC cells [151]. This UCNP nanostructure was based on a multifunctional Y_1_Rs-targeting ligand [Pro30, Nle31, Bpa32, Leu34]NPY(28–36), abbreviated to NPY and loaded with merocyanine 540 (MC540) PS to form LiLuF4:Yb,Er@nLiGdF4@mSiO_2_ (MNPs) [151]. Then 1,2-distearoyl-sn-glycero-3-phosphoethanolamine-N-carboxy (polyethylene glycol)—2000] (DSPE–PEG) was coated onto the surface of MNPs to improve water solubility and biocompatibility of the final nanocomposite [151]. Within 980 nm in vitro PDT assays BC MCF-7 cells treated with MNPs(MC540) noted a 84.8% cell death, whereas those treated with MNPs(MC540)/DSPE-PEG reported a 86.7% cell death and those treated with MNPs(MC540)/DSPE-PEG-NPY noted a 93.5% cell death, suggestive that active targeting and uptake was present (Figure 11) [151]. Within 980 nm in vivo PDT assays on MCF-7 BC induced female BALB/c nude mice tumor volumes of the groups treated with MNPs(MC540), MNPs(MC540)/DSPE-PEG, and MNPs(MC540)/DSPE-PEG-NPY increased over the first 4 days and then decreased on the sixth day after a double PDT application was performed [151]. After 28 days of a double PDT application, mice injected with MNPs(MC540)/DSPE-PEG-NPY showed no tumor growth [151].

In a study performed by Ramírez-García et al. (2018), a UCNP nanoconjugate was constructed for the targeted PDT and imaging against HER-2-positive BC cells, as well as to try and overcome the limited tumor cell depth penetration visible light has [152]. NaYF4:Yb,Er UCNPs were fabricated and attached to a zinc tetracarboxyphenoxy phthalocyanine (ZnPc) PS and a trastuzumab (Tras) HER-2 specific monoclonal antibody to form a UCNPs-ZnPc-Tras nanocomposite [152]. The covalent bonds between UCNPs and ZnPc resulted in resonance energy transfer from the NPs to the PS, which in turn produced cytotoxic singlet oxygen and higher ^1^O_2_ quantum yields when compared to control groups [152]. The PDT efficacy of this nanocomposite was evaluated in vitro within HER-2 positive SK-BR-3 and HER-2 negative MCF-7 human BC cells [152]. Cytotoxicity assays post-PDT at 975 nm noted higher values in HER-2 positive SK-BR-3 BC cells than when compared to HER-2 negative MCF-7 human BC cells, suggestive that enhanced PS targeting uptake was present due to specific HER-2 targeting [152]. Moreover, post-PDT HER-2 positive SK-BR-3 BC cells noted a 93.5% cell death, when compared to HER-2 negative MCF-7 human BC cells which reported a mere 21.8% cell death, suggestive that this nanocomposite was capable of specific and far more enhanced HER-2 positive BC receptor mediated targeted PDT [152].

As previously mentioned, photocatalysis TiO_2_ NPs are nontoxic and have a high photochemical stability to yield improved levels of ROS upon irradiation [153]. When TiO_2_ NPs are utilized within PDT applications as PS, a far higher and controlled loading with improved uptake has been reported [154]. Furthermore, other studies noted that when doping metal atoms, such as ZrO_2_, are attached to TiO_2_ heterostructures, they can temporarily constrain the high recombination rate of photogenerated electron–hole pairs in TiO_2_ NPs when electron–hole pairs migrate from the inside of the photocatalyst to the surface, improving PDT treatment outcomes [154,155]. In a more recent study performed by Ramírez-García et al. (2019), a NaYF4:Yb,Tm UCNP core was fabricated and coated with photo-effecting material TiO_2_-ZrO_2_ as a shell to improve NIR-triggered PDT (NaYF4:Yb,Tm@TiO_2_/ZrO_2_ core@shell NPs) [153]. The monoclonal antibody known as Tras was also added to the UCNPs surface, to improve its overall NP PS active targeting within HER-2 positive in vitro cultured SK-BR-3 human BC cells [153]. Within PDT assays at 975 nm irradiation at 400 μg/mL the NaYF4:Yb,Tm@TiO_2_/ZrO_2_–tras nanocomposite reported 76% cell death, whereas control groups treated with single TiO_2_ UCNP that lacked ZrO_2_ attachment, only 40% cell death was found [153]. Overall, these results revealed that the combinative photocatalytic activity of TiO_2_–ZrO_2_ within the final nanocomposite, improved the PDT treatment outcomes in BC cells due to higher levels of ROS being produced [153].

Studies by Feng et al. (2019) employed a promising strategy called a “all-in-one”, whereby imaging and therapeutic PDT functions were integrated into one nanoplatform, by anchoring a PSs to UCNPs to allow for dual imaging-guided PDT within in vitro MCF-7 BC cells [156]. A bioorthogonal chemical reaction was utilized in this study to allow for a “off/on” state of PDT, in order to circumvent any issues associated with photoactivity of preloaded norbornene-rose bengal (RB-NB) PS, since it can produce skin photosensitivity and so damage normal cells [156]. Thus, a NaYF4: Er, Yb@NaYF4 UCNP was synthesized and covalently bound to a pre-targeting tetrazine (Tz) and FA molecule to form a UCNPs-Tz/FA-PEG (Figure 12a), which was utilized as the one handle of the bioorthogonal reaction in tracking and imaging of deep-seated tumors, since it lacked PS [156]. Then when the RB–NB PS were attached on the surface of the nanoplatform via a bioorthogonal chemical reaction (as the other handle of the UCNP), it demonstrated efficient PS targeting, UCNP energy transfer to the PS, with high yields of ROS and so enhanced treatment within in vitro BC tumors under 980 nm irradiation (Figure 12b) [156]. Upconversion luminescence (UCL) imaging of the nude mice injected with MCF-7 BC cells showed high accumulation of the nanoplatform in tumor sites, due to FA active targeting and EPR effect [156]. Furthermore, in vivo PDT assays on these tumor bearing mice when treated with NPs-Tz/FA-PEG + RB–NB under 980 nm irradiation provided 75.5% decrease in tumor size when compared to control groups [156]. 

A lot of research has been carried within the utilization of UCNPs for the enhanced PS delivery and PDT treatment of BC, due to their ability to be allow for PS activation within the higher NIR wavelength ranges, and consequently be able to provide deeper penetration of tumor tissues, when compared to visible light applications, since the upconversion visible emission from UCNPs can excite PS to produce more ROS [140,157]. It is envisaged that the integration of UCNPs with more NIR penetrable light and idealistic PSs will potentiate near-future targeted PDT BC clinical trials.

## 8. Conclusions and Perspectives

BC is invasive form of cancer, which can metastasize, and frequently recurs after treatment [2]. Many conventional therapies utilized for BC often present themselves with some form of resistance and unwanted side-effects, and surgery is invasive [3]. In this sense, actively targeted PDT is gaining a prominent position as a non-invasive, limited side effect approach for the treatment of BC.

The combination of NPs with PSs, to passively, as well as actively enhance their accumulation in tumor tissues more selectively in order to enhance PDT treatment outcomes, as well as lessen the unwanted side effects on localized tissues is fast becoming a popular approach [22,23].

Inorganic NPs have unique properties, which assist in reducing PS leaching, allow for a high loading capacity of PSs, improve PS passive uptake via the EPR effect, and allow for ease of functionalization with various ligands to promote active PS absorption and, thus, allow for the overall enhancement of PDT BC treatment [27,158]. Furthermore, inorganic and metallic PS nanocarriers are less susceptible to degradation and do not release attached PSs, but rather allow activated ROS after irradiation to diffuse out of them, when compared to organic NPs, and so are more prominently utilized within the field of PDT [29,158].

Gold NPs for example have shown surface plasmon resonance effects that can intensify singlet oxygen quantum yield, as well as induce hyperthermia promoting the overall effect of PDT [28]. Furthermore, inorganic NPs, such as UCNPs can provide a deeper penetration of light in tumors [140] or porous silica NPs allow for the entrapment of oxygen to improve overall PDT treatment outcomes [28].

Anchoring of active targeting moieties to PS-loaded inorganic NPs, allow nanosystems to be specifically directed towards BC cells only, allowing enhanced PS accumulation, which is localized in tumor target cells, only limiting unwanted side effects on normal cells [22,23]. It is also noteworthy to emphasize that the number of receptors per tumor cell is 10^5^, while the number of the PS molecules that can be attached to an inorganic NP to obviate cancer cells is 10^7^, allowing each tumor receptor to be able to at least receive a 10^2^ PS concentration [159]. Thus, the binding of targeting ligand to PS-loaded NPs is imperative to ensure the highest uptake possible of PSs in tumor cells, in order to promote PDT treatment outcomes [159].

It is postulated that, in the near future, the applications of nanotechnology to potentiate PDT should allow for the widespread of breast cancer amongst women to be overcome [159]. However, additional comprehensive studies are still required to scrutinize the physiochemical, pharmacokinetic properties, and safety profiles of nanocarriers, so that maximum accumulation and PS uptake can be attained in the target tissues. In addition, although anchoring of the PSs on the surface of NPs can enhance their biocompatibility, the potential toxic effects and unwanted liver and renal accumulation, must be taken into consideration. Thus, it is imperative that the above discussed and reviewed inorganic NP studies for the actively applied targeted PDT treatment of BC in vitro and in vivo be investigated further within clinical trials, so that the possible future targeted PDT treatment of BC can become a reality.

## Figures and Tables

**Figure 1 pharmaceutics-13-00296-f001:**
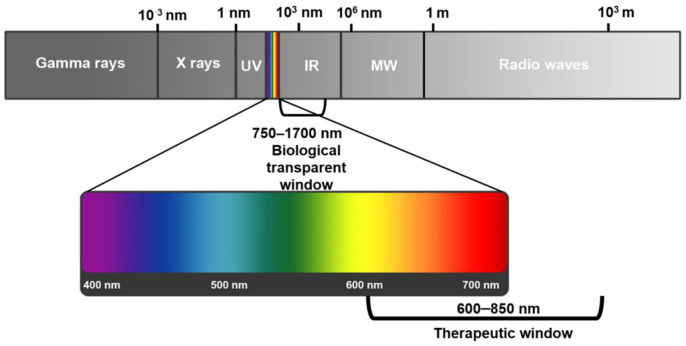
Electromagnetic spectrum showing the ideal phototherapeutic window for photodynamic therapy (PDT) treatment of cancer.

**Figure 2 pharmaceutics-13-00296-f002:**
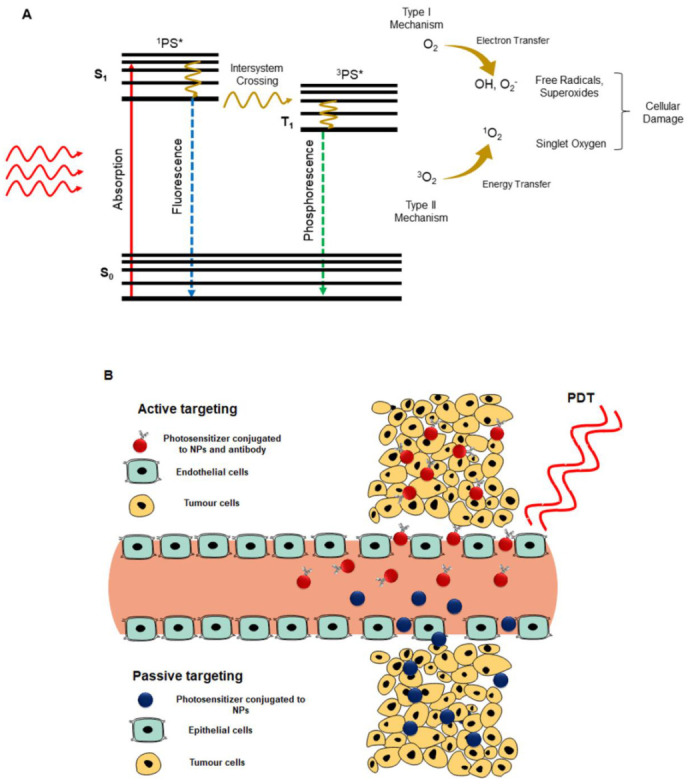
(**A**) PDT mechanism of action, as well as (**B**) passive and active tumor photosensitizer (PS) targeting approaches to generate reactive oxygen species (ROS) and singlet oxygen for tumor destruction (PS* indicates an excited state photosensitizer).

**Figure 3 pharmaceutics-13-00296-f003:**
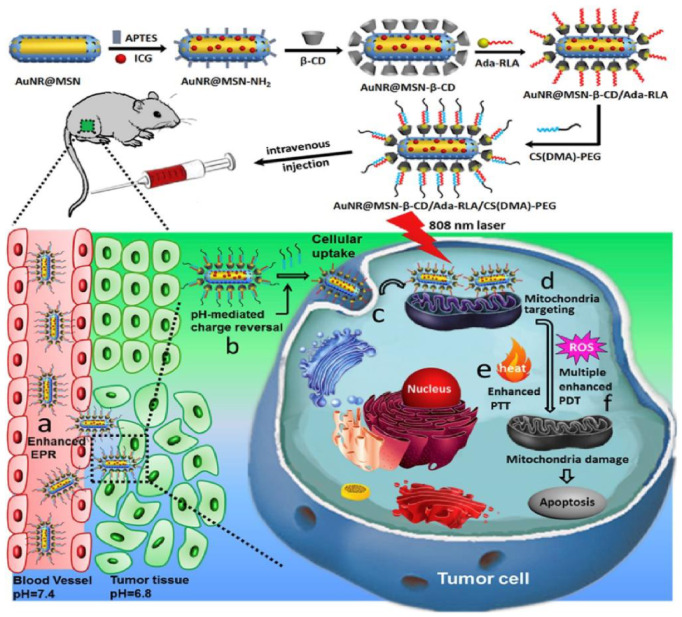
Schematic representation of multifunctional nanoplatform AuNR@MSN and its in vivo process. Reprinted with permission from reference [51] Copyright 2018 Elsevier.

**Figure 4 pharmaceutics-13-00296-f004:**
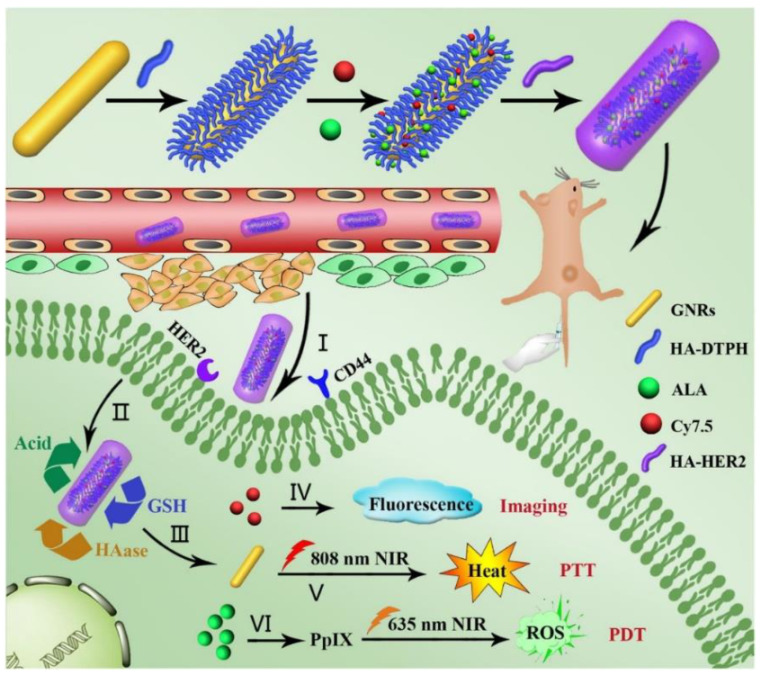
Schematic illustration for preparing GNR-HA^-ALA/Cy7.5^-HER2 with triple-responsive drug release and its application for HER2/CD44 dual-targeted and fluorescence imaging-guided combined PDT/ photothermal therapy (PTT) treatment of breast cancer. Reprinted with permission from reference [57]. Copyright 2019 Elsevier.

**Figure 5 pharmaceutics-13-00296-f005:**
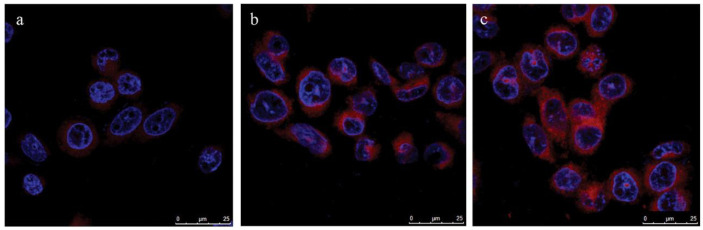
Cellular uptake of (**a**) free PS, (**b**) Fe_3_O_4_-Au-LA-PS (MGPS) and (**c**) Affibody -MGPS.by SK-OV-3 cells. Reprinted with permission from reference [67]. Copyright 2014 Elsevier.

**Figure 6 pharmaceutics-13-00296-f006:**
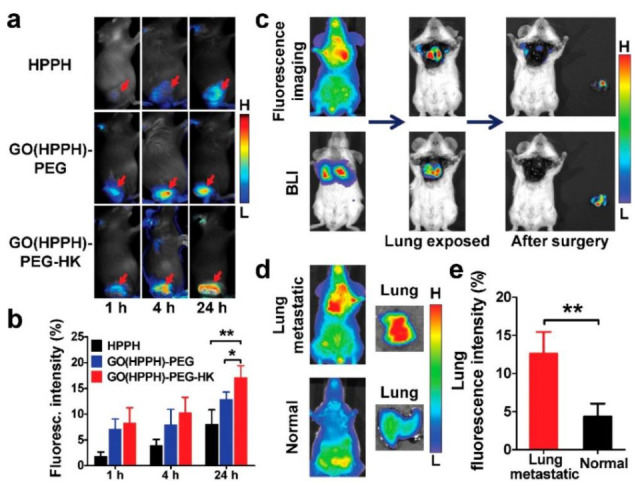
(**a**) Optical images of 4T1 tumor-bearing BALB/c mice, (**b**) quantitative analysis uptake by 4T1 tumors after injection of HPPH, GO(HPPH)-PEG, or GO(HPPH)-PEG-HK, (**c**) In vivo optical imaging and BLI of 4T1-fLuc tumor-bearing BALB/c mice after injection of GO(HPPH)-PEG-HK. (**d**,**e**) Optical images and quantitative analysis of lung uptake of GO(HPPH)-PEG-HK by 4T1-fLuc tumor-bearing and normal BALB/c mice at 24 h post-injection. Reprinted with permission from reference [83]. Copyright 2017 American Chemical Society. * *p* < 0.05, ** *p* < 0.01.

**Figure 7 pharmaceutics-13-00296-f007:**
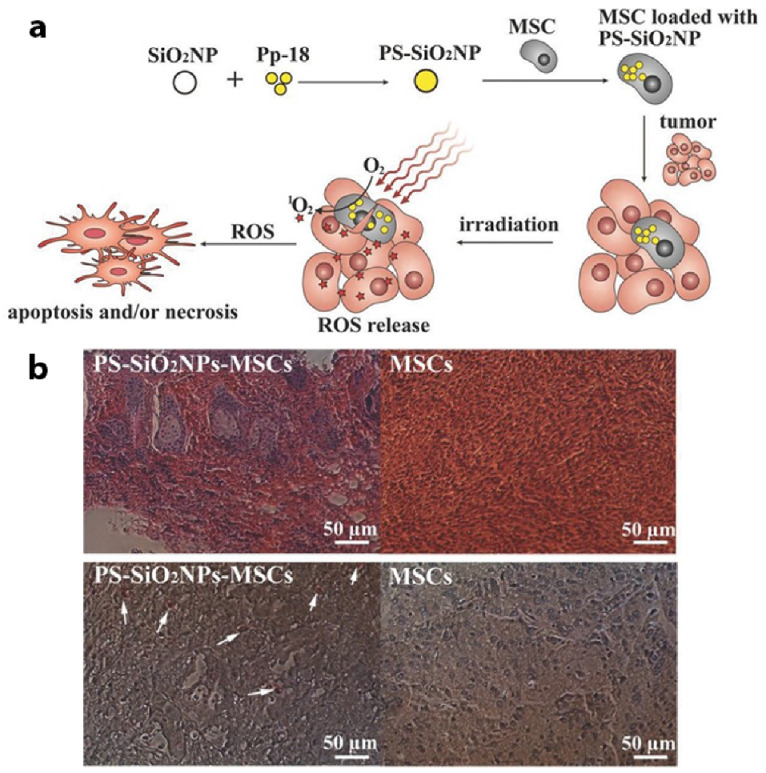
(**a**) PDT treatment of cancer cell using PS-loaded SiO_2_NPs into MSCs, (**b**) In vivo PDT treatment on tumors one day after co-injection of MCF-7 cancer cells and MSCs with (group 1: PS-SiO_2_NPs-MSCs group) or without (group 2: control MSCs group) PS-SiO_2_NPs loaded. Reprinted with permission from reference [114]. Copyright 2014 Willey Online Library.

**Figure 8 pharmaceutics-13-00296-f008:**
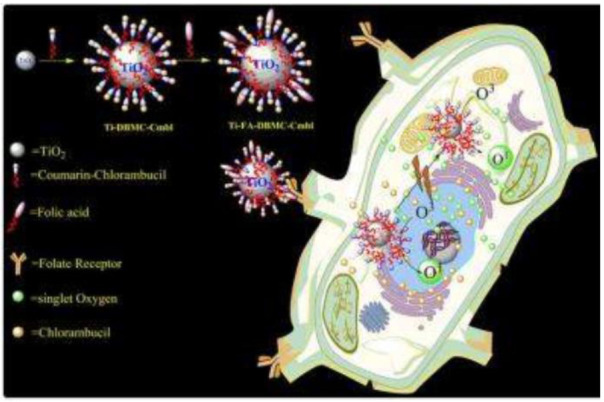
PDT and chemotherapeutic effects of Ti-FA-DBMC-Cmbl NPs on MDA-MB BC cells. Reprinted with permission from reference [129]. Copyright 2015 Royal Chemical Society.

**Figure 9 pharmaceutics-13-00296-f009:**
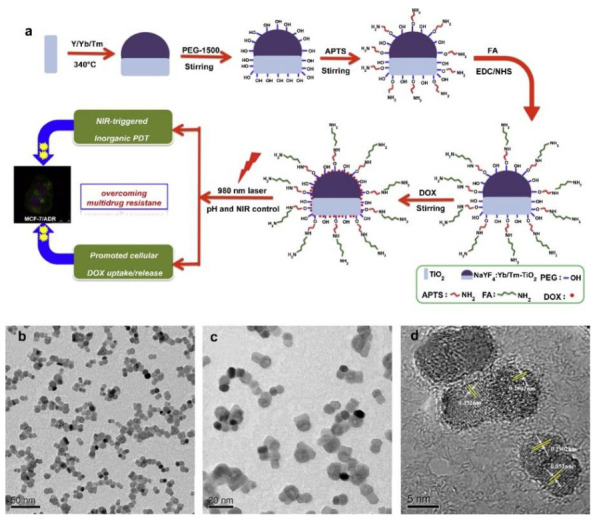
(**a**) Synthesis of DOX-loaded, FA-targeted NaYF4:Yb/TmeTiO_2_ nanocomposites for NIR-triggered PDT and chemotherapy in resistant breast cancer, (**b**–**d**) TEM and HRTEM images of NaYF4:Yb/TmeTiO_2_ nanocomposites. Reprinted with permission from reference [147]. Copyright 2015 Elsevier.

**Figure 10 pharmaceutics-13-00296-f010:**
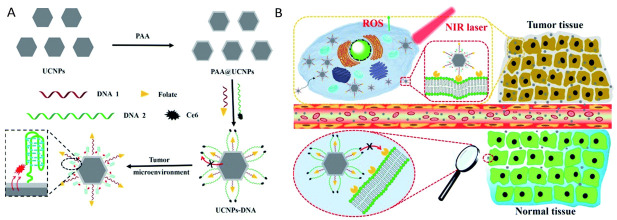
(**A**) Schematic preparation of UCNPs@PAA–DNA and (**B**) specific tumor targeting for PDT treatment of MCF-7 cells [148]. Published by The Royal Society of Chemistry (RSC).

**Figure 11 pharmaceutics-13-00296-f011:**
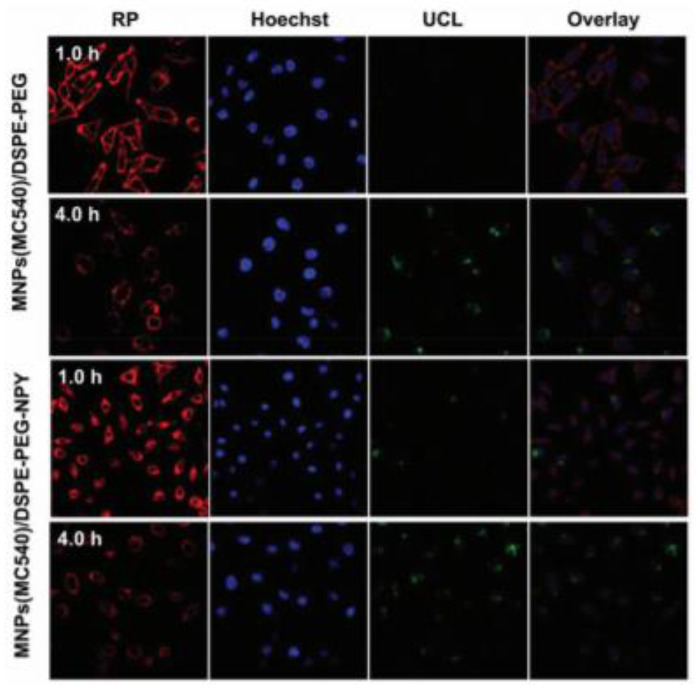
Confocal laser scanning microscopy of MCF-7 incubated with MNPs(MC540)/DSPE-PEG-NPY or MNPs(MC540)/DSPE-PEG. Reprinted with permission from reference [151]. Copyright 2018 Royal Chemical Society.

**Figure 12 pharmaceutics-13-00296-f012:**
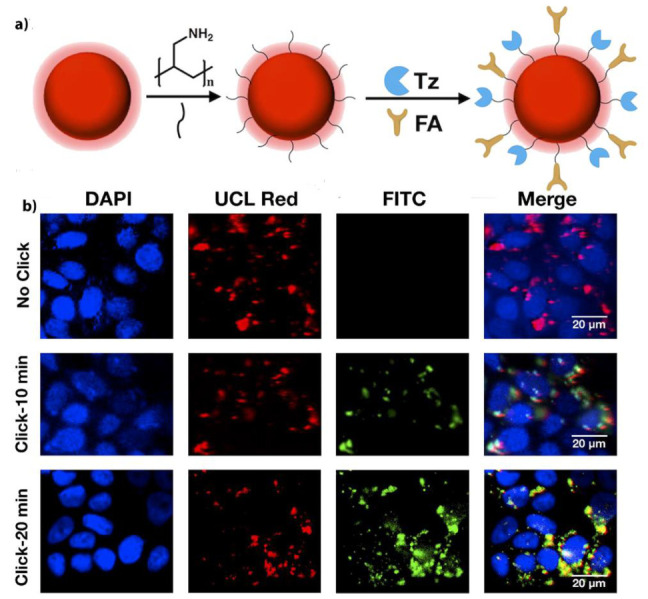
(**a**) Schematic synthesis of UCNPs-Tz/FA-PEG, (**b**) MCF-7 cells treated with UCNPs-Tz/FA-PEG or UCNPs-FA-PEG for UCL imaging with 980 nm light excitation (red channel), and click reaction with FITC-NB (green channel), and stained with DAPI (blue channel). Top: control (no click), middle: click for 10 min and bottom: click for 20 min. Reprinted with permission from reference [156]. Copyright 2019 Elsevier.

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
