# Peer review of "Inorganic Nanoparticles Applied for Active Targeted Photodynamic Therapy of Breast Cancer"

_pharmaceutics, 2021, doi:10.3390/pharmaceutics13030296_

Round 1

Reviewer 1 Report

This review describes the use of nanoparticles in the context of PDT. It is reasonably comprehensive and provides a well-written and clear introduction to the field, so is suitable for publication after some minor revisions. I urge the authors to consider the following relatively minor points:

(a) In Figure 1, I question the use of 650-850 nm as the therapeutic window given a 625 nm laser is usually used for Photofrin, the first photosensitizer dye to find commercial use in the context of PDT.

(b) I recommend that the Jablonski diagram component of Figure 2 is separated out and drafted to a higher quality with more details of the various possible processes provided.

(c) The gold NPs subsection should be converted to noble metal NPs, since other elements such as Ag and Pt are also potentially significant in this context. A focus primarily on AuNPs could still be retained, so only minor changes are needed to accomodate this.

(d) A Jablonski diagram showing the processes involved should be added to the UCNPs subsection along with a slightly more detailed explanation of the photophysical processes involved .

(e) I recommend the use of Chemdraw with ACS settings for the various structures in Table 1.

(f) The BODIPY entry in Table 1 is somewhat misleading since structural modification (e.g. azaBODIPYs, 3,5-distyrylBODIPYs) are usually involved that shifts the main spectra band from the 494-560 nm range. It also particularly important to make it clear that heavy atoms usually have to be introduced to achieve high singlet oxygen quantum yields. I would recommend multiple BODIPY related entries be added to address this in a similar manner to what has been done with the various porphyrin and Pc derivatives.

(g) The spelling of chlorins in Table 1 should be corrected. It currently appears as "chlorines".

Author Response

Reviewer 1

This review describes the use of nanoparticles in the context of PDT. It is reasonably comprehensive and provides a well-written and clear introduction to the field, so is suitable for publication after some minor revisions. I urge the authors to consider the following relatively minor points:

  • In Figure 1, I question the use of 650-850 nm as the therapeutic window given a 625 nm laser is usually used for Photofrin, the first photosensitizer dye to find commercial use in the context of PDT.
  • This statement and its reference have been modified in page 2 line 78-83. Furthermore, Figure 1 has been modified to 600-850 nm.
  • I recommend that the Jablonski diagram component of Figure 2 is separated out and drafted to a higher quality with more details of the various possible processes provided.
  • The Jablonski diagram has been separated from Figure 2 and now it is Figure 2A. More details have been also added to this figure to explain different photophysical processes.
  • The gold NPs subsection should be converted to noble metal NPs, since other elements such as Ag and Pt are also potentially significant in this context. A focus primarily on AuNPs could still be retained, so only minor changes are needed to accommodate this.
  • Thank you so much for your suggestion. We have changed the subsection to noble metal NPs.
  • A Jablonski diagram showing the processes involved should be added to the UCNPs subsection along with a slightly more detailed explanation of the photophysical processes involved.
  • We would rather keep the diagram in section 4 and 5 as the details of the processes have been explained in these sections.
  • I recommend the use of Chemdraw with ACS settings for the various structures in Table 1.
  • All structures in Tables S1 have been modified to ACS setting format.
  • The BODIPY entry in Table 1 is somewhat misleading since structural modification (e.g. azaBODIPYs, 3,5-distyrylBODIPYs) are usually involved that shifts the main spectra band from the 494-560 nm range. It also particularly important to make it clear that heavy atoms usually have to be introduced to achieve high singlet oxygen quantum yields. I would recommend multiple BODIPY related entries be added to address this in a similar manner to what has been done with the various porphyrin and Pc derivatives.
  • The following sentences have been added to the footnote of Table S1 together with references #3 and #4 in order to clarify the activation wavelength range of BODIPY.
  • In order to improve the potential applications of BODIPY in photodynamic cancer therapy, they have been conjugated to metal complexes, heavy atoms and receptor ligands or modification is provided around the core structure to facilitate singlet oxygen generation and extend the activation wavelength range.

(g) The spelling of chlorins in Table 1 should be corrected. It currently appears as "chlorines".

  • The spelling of chlorins has been rectified in Table S1.

Reviewer 2 Report

  1. Line 9: Wavelength of light it, should replace with wavelength of light is
  2. within the 650-850 nm range, should replace within the range of 650-850 nm
  3. Authors should synchronize figure in text either figure of Fig.
  4. Line 133, targeting, most definitely does provide a more selective, does should be dose.
  5. Line 135-36, so more often than not enhanced PDT outcomes, with encouraging cancer treatment results are often observed, Authors have to rewrite in its current form it's not understandable
  6. Line 169, that allows for functionalization with a various of ligands for active targeting [32], need to rewrite.
  7. LIne 175, Gold NPs can also be employed for imaging contrast agents and radiosensitizers, due to the high atomic number gold has [30], need to rewrite
  8. Line 506, compared on a 1:1 GQD:MB ratio [92]. It should replace with compare to. 

Author Response

Reviewer 2

  1. Line 9: Wavelength of light it, should replace with wavelength of light is
  • “Wavelength of light it” has been modified to “wavelength of light is”.
  1. within the 650-850 nm range, should replace within the range of 650-850 nm
  • The statement has been replaced to “within the range of 650-850 nm”. Furthermore, the wavelength range in the manuscript and Figure 1 have been modified to 600-850 nm based on 1st reviewer’s comment.
  1. Authors should synchronize figure in text either figure of Fig.
  • We have synchronized all figures in the manuscript to “Figure”.
  1. Line 133, targeting, most definitely does provide a more selective, does should be dose.
  • “Dose” in this statement is correct and emphasizes on more selectivity of nanoparticle active targeting compared to passive targeting.
  1. Line 135-36, so more often than not enhanced PDT outcomes, with encouraging cancer treatment results are often observed, Authors have to rewrite in its current form it's not understandable
  • The sentences have been modified to the following sentence to clarify it.
  • so higher accumulation of the nanocarrier and cellular concentration of the drug into the cells will take place.
  1. Line 169, that allows for functionalization with a various of ligands for active targeting [32], need to rewrite.
  • The sentences have been modified to the following sentence to clarify it.
  • that can be functionalized with a variety of ligands for active targeting.
  1. Line 175, Gold NPs can also be employed for imaging contrast agents and radiosensitizers, due to the high atomic number gold has [30], need to rewrite
  • The sentences have been modified to the following sentence to clarify it.
  • Gold NPs can be employed for imaging contrast agents and radiosensitizers thanks to the high atomic number of gold.
  1. Line 506, compared on a 1:1 GQD:MB ratio [92]. It should replace with compare to. 
  • “Compared on” has been replaced with “compared to”.